DATA RELEASE

# Spatial and temporal distribution of *Culex* and *Aedes* mosquitoes in Ghana

Yaw Akuamoah-Boateng[1], Christopher Mfum Owusu-Asenso[1],
Anisa Abdulai[1], Abdul Rahim Mohammed Sabtiu[1], Isaac Kwame Sraku[1],
Sebastian Kwo Egyin Mensah[1], Faustina Adobea Owusu[1],
Abena Ahema Ebuako[1], Godfred Amoateng[1], Bright Churchill Obeng[1],
Richard Tettey Doe[1], Emmanuel Nana Boadu[1], Akua Aboagyewaa Appiah[1],
Grace Arhin Danquah[1], Nutifafa Efui Abusah[1],
Dhikrullahi Bunkunmi Shittu[1], Gabriel Akosah-Brempong[2],
Cosmos Manwovor-Anbon Pambit Zong[2], Daniel Kodjo Halou[3],
Osei Kwaku Akuoko[4], Cornelia Appiah-Kwarteng[5] and Yaw Asare Afrane[1],*

1  Centre for Vector-Borne Disease Research, Department of Medical Microbiology, University of Ghana
   Medical School, Korle-Bu, Accra, Ghana
2  African Regional Postgraduate Programme in Insect Sciences (ARPPIS), Department of Animal Biology
   and Conservation Science, College of Basic and Applied Sciences, University of Ghana, Accra, Ghana
3  Department of Vector Biology, Liverpool School of Tropical Medicine, UK
4  Department of Parasitology, Noguchi Memorial Institute for Medical Research, College of Health
   Sciences, University of Ghana, Legon, Accra, Ghana
5  School of Veterinary Medicine, University of Ghana, Legon, Accra, Ghana

**Submitted:** 14 September 2025

\* Corresponding author. E-mail:
yafrane@ug.edu.gh

Preprint submitted at
https://africarxiv.ubuntunet.net/
handle/1/10397

Included in the series: *Vectors of
human disease* (https://doi.org/10.
46471/GIGABYTE_SERIES_0002)

## ABSTRACT

In Africa, *Culex* is an important vector that transmits West Nile virus, whilst *Aedes* mosquitoes transmit dengue, yellow fever, chikungunya, and Zika. However, very limited data is available on their bionomics and ecology. Here, we provide data on the abundance and distribution of *Culex* and *Aedes* mosquitoes in Ghana between 2017 and 2025. We collected 39,761 *Culex* and 6,047 *Aedes* mosquitoes using various mosquito-trapping tools. Both vectors were predominantly observed outdoors. *Aedes aegypti* was the most dominant *Aedes* vector observed in Ghana. The invasive *Aedes albopictus* was sampled in 2023, whereas *Aedes vittatus* was observed in Accra. Our data provides important information to support vector surveillance, ecological risk assessments, and integrated vector-management strategies.

**Subjects**  Ecology, Biodiversity, Taxonomy

## DATA DESCRIPTION
### Background and context

*Culex* mosquitoes are important vectors of emerging arboviruses such as West Nile virus and Japanese encephalitis [1, 2]. Also, *Aedes* mosquitoes are the primary vectors of dengue, Zika, chikungunya, and yellow fever viruses, posing a significant public health threat globally [3]. Invasion of *Aedes albopictus* mosquitoes in the West African sub-region has coincided with multiple outbreaks of dengue fever [4, 5]. However, the distribution of *Culex* and *Aedes* mosquitoes and their role in transmitting these pathogens remains understudied.

This gap limits our ability to map the distribution of *Culex* and *Aedes* species across varying ecological zones to plan targeted vector-control interventions and management strategies. To help bridge this gap, it is important to study the ecology and bionomics of *Culex* and *Aedes* mosquitoes at fine and operational spatiotemporal scales.

In this paper, we present a dataset on the spatial and temporal distribution and ecology of *Culex* and *Aedes* mosquitoes. This resource may guide the targeted control and elimination of arboviral vectors in Ghana. Also, it will help identify potential hotspots for *Culex*- and *Aedes*-borne diseases in relation to urbanization, environmental change, and human exposure.

## METHODS

### Study sites

Mosquito sampling was conducted in twenty-two communities in different ecological settings in Ghana, comprising the Coastal Savannah (Accra, Ada, Aflao, Apam, Dawurampong, Dodowa, Elubo, Medie, Oyarifa, Takoradi, Tema, Anaji, Apowa, Fijai), Forest (Konongo, Twifo Praso, Abetifi, Amedzofe), and Sahel Savannah (Kpalsogu, Pagaza, Libga, Galenkpegu) (Figure 1). The Coastal Savannah is in the south of Ghana and it experiences a bimodal rainfall pattern. The Coastal savannah rainy seasons occur between April and June, and between September and October. The dry season in the Coastal Savannah typically runs from November to March. The Forest zone in the middle of Ghana also experiences a bimodal rainfall pattern with rainy periods between March and June, and between September and October, with a dry season between November and February. Finally, the Sahel Savannah in the north of Ghana has a unimodal rainfall pattern from May to November, whereas the dry season (December–April) sees temperatures rise as high as 42 °C.

### Mosquito sampling methods

Mosquito collections were performed at two time points annually (Dry and Rainy seasons) from 2017 to 2025. Resting adult *Culex* and *Aedes* mosquitoes were collected using Prokopack aspirations, Pyrethrum Spray Catches, Clay pots, and Pit traps [6]. Host-seeking adult *Culex* and *Aedes* mosquitoes were collected using Human Landing Catches (HLC) [7], Biogent Sentinel traps (BG) and Light traps (CDC miniature light trap). At each sampling site, sixteen houses were randomly selected for HLC, BG traps, and light trap collections. Twenty houses were randomly selected for Prokopack aspirations. Samples were collected both indoors and outdoors. Clay pot traps and Pit trap collections were carried out as described by Odiere *et al.* [6] and Mawejje *et al.* [8], respectively. Four houses were randomly selected for clay pot collection, and two pit traps per community.

### Data validation and quality control

Collected mosquito samples were morphologically sorted and identified as *Culex* or *Aedes* mosquitoes. *Aedes* mosquitoes were morphologically identified using well-established morphological keys [9]. *Culex* mosquitoes were not identified to species. Data was thoroughly screened and automatically validated using the Integrated Publishing Toolkit of Global Biodiversity Information Facility (GBIF) [10].



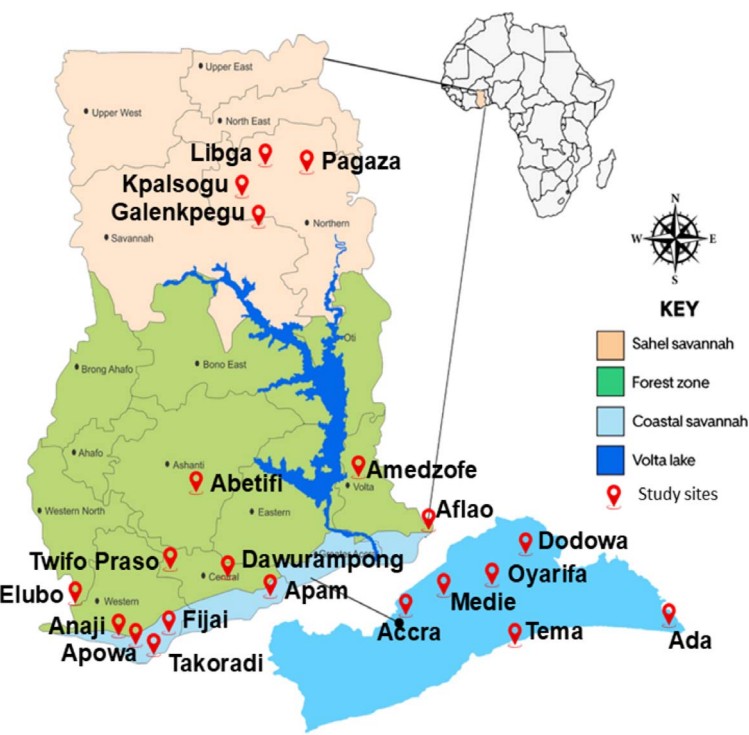

**Figure 1.** Map of Ghana showing the study sites.

## RESULTS

### Abundance and distribution of *Culex* mosquitoes in Ghana

A total of 39,761 *Culex* mosquitoes were collected across the study sites in different ecological zones of Ghana (Table 1). *Culex* mosquito abundance was higher in the rural communities (55.47% of all our collected Culex mosquitoes; 22,056/39,761) than in the urban (34.71%; 13,801/39,761) and peri-urban (9.42%; 3,904/39,761) communities. A high proportion of the *Culex* mosquitoes collected were caught outdoors (80.72%; 32,094/39,761), whereas 19.28% (7,667/39,761) were caught indoors. A Chi-square test of independence showed a statistically significant difference in vector collection between indoor and outdoor sites across the urban, rural, and peri-urban sites ($X^2 = 5800$, $df = 2$, $p < 0.001$).

 *Culex* mosquito abundance was higher during the dry season (60.69%; 24,131/39,761) compared to the rainy season (39.31%; 15,630/39,761) (Figure 2). A statistically significant difference in the seasonal distribution of *Culex* mosquitoes was observed across the urban, rural, and peri-urban sites (Chi-square test, $X^2(2) = 103.20$, $p < 0.001$).

### Abundance and distribution of *Aedes* mosquitoes in Ghana

A total of 6,047 *Aedes* mosquitoes were collected across sites in different ecological zones of Ghana (Table 2). The *Aedes* mosquitoes were sampled predominantly outdoors (88.03%; 5,323/6,047), whereas 11.97% (724/6,047) were sampled indoors. *Aedes aegypti* mosquitoes were the most abundant *Aedes* sampled in all communities (5,817/6,047). *Aedes vittatus* mosquitoes (108/6,047) were found in Accra. *Aedes albopictus* (116/6,047) and *Aedes chemulpoensis* (6/6,047) mosquitoes were found in Takoradi, an urban settlement (Figure 3).



**Table 1.** *Culex* mosquito distribution across Ghana.

| Sampling site | Site classification | Ecological zone | Location | | Total (*n*) |
|---|---|---|---|---|---|
| | | | Indoor (*n*) | Outdoor (*n*) | |
| Abetifi | Peri-urban | Forest | 0 | 59 | 59 |
| Accra | Urban | Coastal | 0 | 12,939 | 12,939 |
| Ada | Rural | Coastal | 5,148 | 8,579 | 13,727 |
| Aflao | Urban | Coastal | 429 | 192 | 621 |
| Amedzofe | Rural | Forest | 0 | 2 | 2 |
| Apam | Rural | Coastal | 41 | 36 | 77 |
| Dawurampong | Rural | Coastal | 3 | 1 | 4 |
| Dodowa | Rural | Coastal | 732 | 1,562 | 2,294 |
| Elubo | Urban | Coastal | 15 | 55 | 70 |
| Galenkpegu | Rural | Sahel-savannah | 10 | 2 | 12 |
| Konongo | Rural | Forest | 188 | 650 | 838 |
| Kpalsogu | Rural | Sahel-savannah | 475 | 1,333 | 1,808 |
| Libga | Rural | Sahel-savannah | 256 | 1,079 | 1,335 |
| Medie | Peri-urban | Coastal | 0 | 2,478 | 2,478 |
| Oyarifa | Peri-urban | Coastal | 0 | 1,297 | 1,297 |
| Pagaza | Rural | Sahel-savannah | 320 | 1,443 | 1,763 |
| Takoradi | Urban | Coastal | 0 | 235 | 235 |
| Tema | Urban | Coastal | 0 | 6 | 6 |
| Twifo Praso | Rural | Forest | 50 | 120 | 170 |
| **Total** | | | **7,667** | **32,094** | **39,761** |

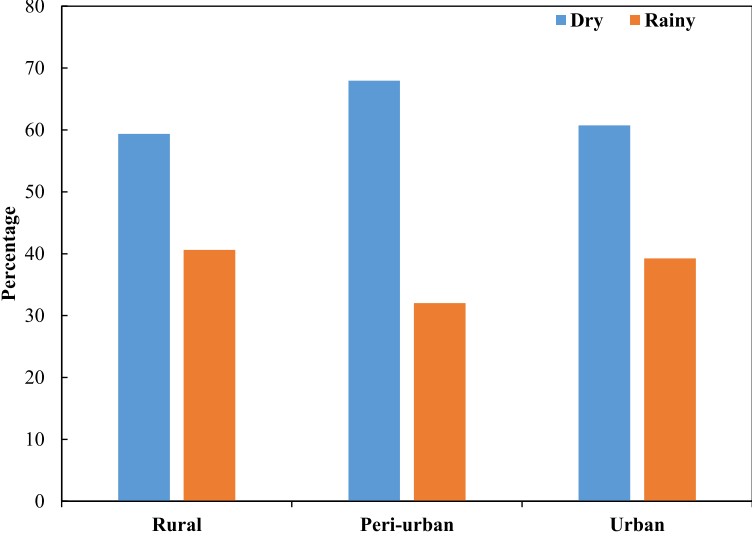

**Figure 2.** Seasonal distribution of *Culex* mosquitoes in Ghana.

The invasive *Aedes albopictus* was first sampled in 2023 [11]. A significant association was observed between the urban-rural continuum and mosquito collection location ($X^2(1) = 989.60$, $p < 0.001$). *Aedes* mosquitoes in the rural areas were more likely to be collected indoors (39.6%), whereas those in the urban areas were mostly collected outdoors (94.3%).

More *Aedes* mosquitoes were collected during the dry season than during the rainy season in all ecological zones. In the rural sites, 62.91% (697/1,108) of *Aedes* mosquitoes were sampled in the dry season, whereas 37.09% (411/1,108) were sampled in the rainy



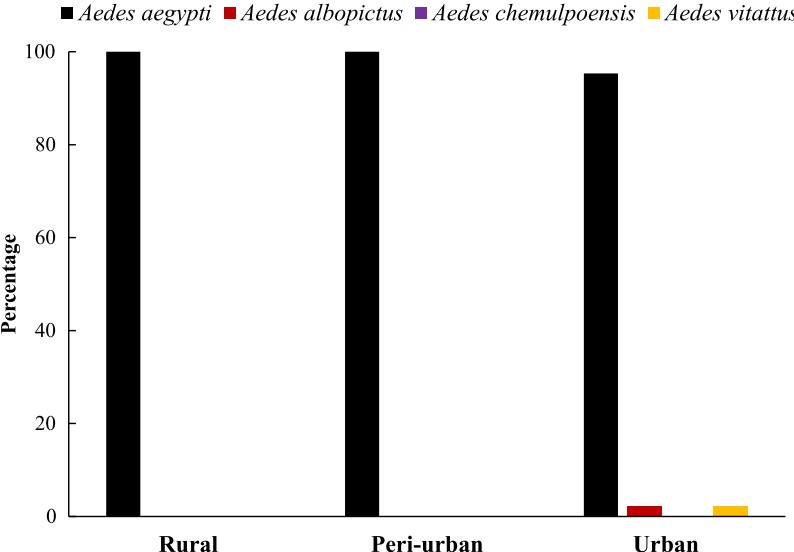

**Figure 3.** *Aedes* mosquito species distribution across Ghana.

**Table 2.** *Aedes* mosquito abundance and distribution across Ghana.

| Site | Site classification | Ecological zone | Location | | Total (*n*) |
|---|---|---|---|---|---|
| | | | Indoor (*n*) | Outdoor (*n*) | |
| Abetifi | Peri-urban | Forest | 2 | 0 | 2 |
| Accra | Urban | Coastal | 129 | 2,695 | 2,824 |
| Ada | Rural | Coastal | 270 | 251 | 521 |
| Aflao | Urban | Coastal | 7 | 19 | 26 |
| Anaji | Urban | Coastal | 0 | 114 | 114 |
| Apam | Rural | Coastal | 31 | 27 | 58 |
| Apowa | Urban | Coastal | 47 | 396 | 443 |
| Dawurampong | Rural | Coastal | 12 | 2 | 14 |
| Dodowa | Rural | Coastal | 17 | 56 | 73 |
| Elubo | Urban | Coastal | 1 | 5 | 6 |
| Fijai | Urban | Coastal | 0 | 1 | 1 |
| Galenkpegu | Rural | Sahel-savannah | 0 | 1 | 1 |
| Konongo | Rural | Forest | 11 | 13 | 24 |
| Kpalsogu | Rural | Sahel-savannah | 9 | 63 | 72 |
| Libga | Rural | Sahel-savannah | 66 | 146 | 212 |
| Pagaza | Rural | Sahel-savannah | 17 | 108 | 125 |
| Takoradi | Urban | Coastal | 70 | 1,231 | 1,301 |
| Tema | Urban | Coastal | 29 | 194 | 223 |
| Twifo Praso | Rural | Forest | 6 | 1 | 7 |
| **Total** | | | **724** | **5,323** | **6,047** |

season. In the peri-urban sites, 100% (2/2) were sampled in the dry season. Similarly, in the urban sites, 63.62% (3,141/4,937) of the *Aedes* mosquitoes caught were sampled during the dry season, whereas 36.38% (1,706/4,937) were caught during the rainy season (Figure 4).

## RE-USE POTENTIAL

The two datasets presented here offer valuable insights into the distribution of *Culex* and *Aedes* mosquito species in Ghana. Researchers and public health professionals can use this

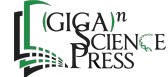

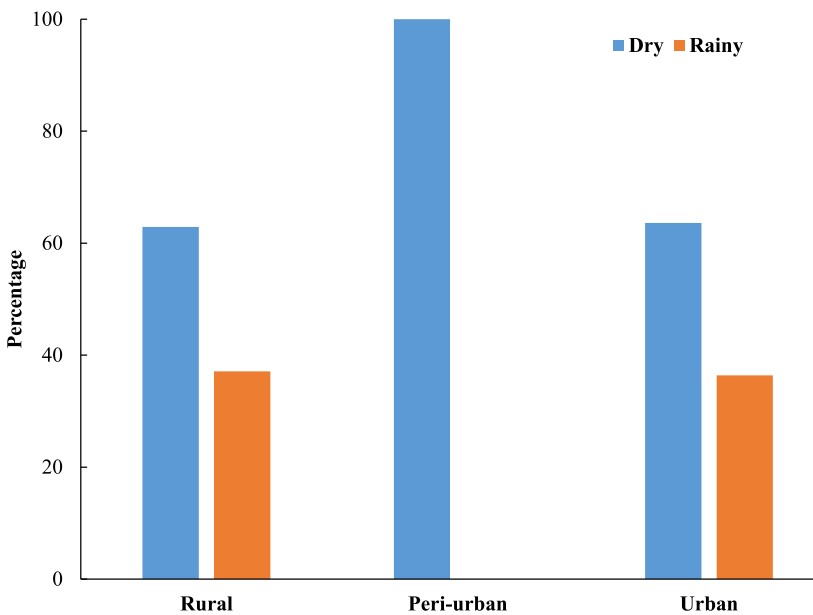

**Figure 4.** Seasonal distribution of *Aedes* mosquitoes in Ghana.

dataset to enhance their understanding of the ecology and distribution of *Culex* and *Aedes* mosquitoes. The information in the dataset can serve as a resource for studies assessing transmission risks, vector control strategies, disease surveillance, and a broader understanding of the ecology of *Culex* and *Aedes* mosquitoes across Ghana's various ecological zones.

## DATA AVAILABILITY

The two datasets described here are available via the GBIF repository [12, 13].

## EDITOR'S NOTE

This paper is part of a series of Data Release articles working with GBIF and supported by TDR, the Special Program for Research and Training in Tropical Diseases, hosted at the World Health Organization [14].

## ABBREVIATIONS

BG, Biogent Sentinel traps; CDC, Centers for Disease Control and Prevention; GBIF: Global Biodiversity Information Facility; HLC, Human Landing Catches.

## DECLARATIONS

### Ethical approval

This study received scientific and ethical approval from the Ethical and Protocol Review Committee of the College of Health Sciences, University of Ghana, Korle-Bu Campus. Informed consent, both verbal and written, was obtained from opinion leaders at the selected study sites as well as from house or household heads prior to conducting mosquito sampling at all study locations.

## Consent for publication

Not applicable.

## Competing interests

The authors declare that they have no competing interests.

## Authors' contributions

YAA conceived, designed and supervised the study. ARM, YAB, CMOA, AA, IAK, SM, FAO, AEA, GA, OBC, RD, ENB, GAD, AAA, DS, GAB, CMPZ, DH and OS were responsible for the study data collection and contributed to the analysis of the data. YAB, CMOA and CAK drafted the manuscript. All the authors read and approved the final manuscript.

## Funding

This study was funded by grants from the National Institutes of Health (R01 A1123074, RO3 AI186018, and D43 TW 011513). The funding agency had no involvement in the study's design, data collection, analysis, interpretation, or manuscript preparation.

## Acknowledgements

We sincerely appreciate the communities that served as study sites for permitting us to conduct sampling within their homes and neighborhoods.

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
