## [Reviewer Report]

Indicate in the comments box below whether you are happy with the changes made or if the manuscript is unacceptable.Comments on revised manuscriptThis manuscript presents a high-quality, valuable dataset on the spatial and temporal distribution of Culex and Aedes mosquitoes in Ghana, collected over eight years using standardized methods, the GBIF data is clear and published; therefore, it is recommended for acceptance.Indicate in the comments box below whether you are happy with the changes made or if the manuscript is unacceptable.Comments on revised manuscriptThis manuscript presents a high-quality, valuable dataset on the spatial and temporal distribution of Culex and Aedes mosquitoes in Ghana, collected over eight years using standardized methods, the GBIF data is clear and published; therefore, it is recommended for acceptance.

---

## [Editor Report]

Editor’s AssessmentIn Africa, Culex is an important vector that transmits West Nile Virus whilst Aedes mosquitoes transmits dengue, yellow fever, chikungunya and zika. There is however very limited data on their bionomics and ecology, hindering modelling research and control strategies.. This paper is one of a series of Data Release papers in GigaByte supported by TDR and the WHO describing datasets hosted in GBIF to tackle these data gaps in vectors of human disease data. This paper presents a dataset on the spatial and temporal distribution and ecology of Culex and Aedes mosquitoes to support targeted arboviral vector control and elimination efforts in Ghana. Mosquito sampling conducted in twenty-two communities in different ecological settings across the country, performed at two time points annually (Dry and Rainy seasons) from 2017 to 2025 using various mosquito trapping tools. A total of 39,761 Culex mosquitoes were collected in the study sites alongside 6,047 Aedes mosquitoes. Peer review and data auditing found the data to be well validated. The information contained can serve as a resource for studies focused on assessing transmission risks, vector control strategies, disease surveillance and a broader comprehension of Culex and Aedes mosquito ecology in the various ecological zones of Ghana.Editor’s AssessmentIn Africa, Culex is an important vector that transmits West Nile Virus whilst Aedes mosquitoes transmits dengue, yellow fever, chikungunya and zika. There is however very limited data on their bionomics and ecology, hindering modelling research and control strategies.. This paper is one of a series of Data Release papers in GigaByte supported by TDR and the WHO describing datasets hosted in GBIF to tackle these data gaps in vectors of human disease data. This paper presents a dataset on the spatial and temporal distribution and ecology of Culex and Aedes mosquitoes to support targeted arboviral vector control and elimination efforts in Ghana. Mosquito sampling conducted in twenty-two communities in different ecological settings across the country, performed at two time points annually (Dry and Rainy seasons) from 2017 to 2025 using various mosquito trapping tools. A total of 39,761 Culex mosquitoes were collected in the study sites alongside 6,047 Aedes mosquitoes. Peer review and data auditing found the data to be well validated. The information contained can serve as a resource for studies focused on assessing transmission risks, vector control strategies, disease surveillance and a broader comprehension of Culex and Aedes mosquito ecology in the various ecological zones of Ghana.

---

## [Reviewer Report]

Upload additional filesDRR-202509-02-R01/stage_files/DRR-202509-02/Review MS/GBIF-Data-Review-DRR-202509-02.docxReviewer name and names of any other individual's who aided in reviewer Yannan FanDo you understand and agree to our policy of having open and named reviews, and having your review included with the published papers. (If no, please inform the editor that you cannot review this manuscript.)YesIs the language of sufficient quality?YesPlease add additional comments on language quality to clarify if needed
Are all data available and do they match the descriptions in the paper? NoAdditional CommentsAre the data and metadata consistent with relevant minimum information or reporting standards? See GigaDB checklists for examples <a href="http://gigadb.org/site/guide" target="_blank">http://gigadb.org/site/guide</a>YesAdditional CommentsThe current GBIF data for Culex and Aedes mosquitoes are not match to the MSIs the data acquisition clear, complete and methodologically sound?YesAdditional CommentsIs there sufficient detail in the methods and data-processing steps to allow reproduction?YesAdditional CommentsIs there sufficient data validation and statistical analyses of data quality? YesAdditional CommentsIs the validation suitable for this type of data?YesAdditional CommentsIs there sufficient information for others to reuse this dataset or integrate it with other data?YesAdditional CommentsAny Additional Overall Comments to the AuthorRecommendationMinor Revision

---

## [Reviewer Report]

Reviewer name and names of any other individual's who aided in reviewer Oscar Alexander Aguirre-ObandoDo you understand and agree to our policy of having open and named reviews, and having your review included with the published papers. (If no, please inform the editor that you cannot review this manuscript.)YesIs the language of sufficient quality?YesPlease add additional comments on language quality to clarify if needed
Regarding disease names, only Zika should be capitalized, as it is a proper noun derived from the Zika Forest in Uganda. All other disease names should remain in lowercaseAre all data available and do they match the descriptions in the paper? YesAdditional CommentsAre the data and metadata consistent with relevant minimum information or reporting standards? See GigaDB checklists for examples <a href="http://gigadb.org/site/guide" target="_blank">http://gigadb.org/site/guide</a>YesAdditional CommentsIs the data acquisition clear, complete and methodologically sound?YesAdditional CommentsIs there sufficient detail in the methods and data-processing steps to allow reproduction?YesAdditional CommentsIs there sufficient data validation and statistical analyses of data quality? NoAdditional CommentsI recommend performing a statistical analysis on the species abundance and distribution data to strengthen the reliability of the results presented.Is the validation suitable for this type of data?YesAdditional CommentsIs there sufficient information for others to reuse this dataset or integrate it with other data?YesAdditional CommentsAny Additional Overall Comments to the AuthorIn the tables, the names of the genera mentioned should be italicized. Additionally, a horizontal line should be inserted before the total row to visually distinguish this value from the rest of the tableRecommendationMinor Revision